

# High phenotypic plasticity at the dawn of the eosauropterygian radiation

Antoine Laboury[1], Torsten M. Scheyer[2], Nicole Klein[3], Thomas L. Stubbs[4] and Valentin Fischer[1]

[1] Evolution & Diversity Dynamics Lab, Université de Liège, Liège, Belgium
[2] Department of Palaeontology, University of Zurich, Zurich, Switzerland
[3] Institute of Geosciences, Paleontology, University of Bonn, Bonn, Germany
[4] School of Life, Health & Chemical Sciences, Open University, Milton Keynes, United Kingdom

## ABSTRACT

The initial radiation of Eosauropterygia during the Triassic biotic recovery represents a key event in the dominance of reptiles secondarily adapted to marine environments. Recent studies on Mesozoic marine reptile disparity highlighted that eosauropterygians had their greatest morphological diversity during the Middle Triassic, with the co-occurrence of Pachypleurosauroidea, Nothosauroidea and Pistosauroidea, mostly along the margins of the Tethys Ocean. However, these previous studies quantitatively analysed the disparity of Eosauropterygia as a whole without focussing on Triassic taxa, thus limiting our understanding of their diversification and morphospace occupation during the Middle Triassic.
Our multivariate morphometric analyses highlight a clearly distinct colonization of the ecomorphospace by the three clades, with no evidence of whole-body convergent evolution with the exception of the peculiar pistosauroid *Wangosaurus brevirostris*, which appears phenotypically much more similar to nothosauroids. This global pattern is mostly driven by craniodental differences and inferred feeding specializations. We also reveal noticeable regional differences among nothosauroids and pachypleurosauroids of which the latter likely experienced a remarkable diversification in the eastern Tethys during the Pelsonian. Our results demonstrate that the high phenotypic plasticity characterizing the evolution of the pelagic plesiosaurians was already present in their Triassic ancestors, casting eosauropterygians as particularly adaptable animals.

# INTRODUCTION

The Triassic biotic recovery following the Permian–Triassic boundary mass extinction (PTME) represents a crucial episode in Earth's history, characterized by the colonization of the oceans by reptiles and the emergence of modern trophic networks in these aquatic ecosystems that are still in place today (*Benton et al., 2013*; *Fröbisch et al., 2013*; *Scheyer et al., 2014*; *Liu et al., 2014*; *Kelley & Pyenson, 2015*; *Huang et al., 2020*; *Sander et al., 2021*). Marine reptiles dominated the whole Mesozoic and explored numerous ecological niches as demonstrated by their ecomorphological diversification (*Stubbs & Benton, 2016*; *Foffa*

Corresponding author
Antoine Laboury,
a.laboury@uliege.be

*et al., 2018*; *Reeves et al., 2021*; *Sander et al., 2021*; *MacLaren et al., 2022*; *Fischer et al., 2022*). These aquatic reptiles experienced an unprecedented burst of diversification during the Middle Triassic, likely driven by the novel ecological opportunities provided by the shallow epicontinental seas connected to the Paleotethys and Panthalassa oceans (*Benson & Butler, 2011*; *Stubbs & Benton, 2016*; *Moon & Stubbs, 2020*; *Reeves et al., 2021*). Sauropterygia is the most speciose and the longest-living (Olenekian–Maastrichtian; *e.g.*, *Benson et al., 2010*; *Jiang et al., 2014*) clade of marine reptiles and its members were key components of marine trophic chains for the entire Mesozoic. This clade is divided into two major lineages, the durophagous Placodontia and the disparate Eosauropterygia which includes the lizard-like pachypleurosauroids, the flat-headed nothosauroids, and the long-necked pistosauroids, in which plesiosaurians are nested (*Rieppel, 2000*; *Motani, 2009*). The Triassic representatives of Sauropterygia are essentially restricted to the western and eastern margins of the Paleotethys (outcropping in present-day Europe and China, respectively) (*Rieppel, 2000*; *Bardet et al., 2014*) even if some taxa such as *Corosaurus* and *Augustasaurus* and remains with nothosauroidean affinity have been found in Eastern Panthalassa as well (outcropping in present-day North America) (*Case, 1936*; *Sander, Rieppel & Bucher, 1997*; *Rieppel, 2000*; *Bardet et al., 2014*; *Scheyer, Neuman & Brinkman, 2019*).

Recent studies of marine reptile disparity through time have demonstrated that sauropterygians became the most disparate clade by the Anisian (*Stubbs & Benton, 2016*; *Reeves et al., 2021*) and that morphological diversity was mostly driven by the emergence of the profound durophagous adaptations of placodonts (*Stubbs & Benton, 2016*; *Reeves et al., 2021*; *Fischer et al., 2022*). Concerning eosauropterygians, qualitative observations in the fossil record reveal a diversification of morphologies related to both their feeding strategies (*Rieppel, 2002*) and swimming modes during the Middle Triassic (*Krahl, Klein & Sander, 2013*; *Klein et al., 2016*; *Xu et al., 2022*). Quantitative analyses suggest a burst in skull size and high disparity during that period (*Stubbs & Benton, 2016*), associated with the appearance of small-sized pachypleurosauroids and gigantic nothosaurians (*Liu et al., 2014*). Post-Triassic sauropterygians (*i.e.*, Plesiosauria) would seemingly never again reach such a high disparity even if their evolution was punctuated by periods of high morphological diversification, craniodental convergences and variations in neck elongation (*Stubbs & Benton, 2016*; *Fischer et al., 2018*, *2020*; *Reeves et al., 2021*).

However, studies which have analyzed the disparity of Sauropterygia mostly consider the clade as a whole, or only investigate the morphological evolution of the derived plesiosaurians, leaving thus the Triassic clades relatively understudied. As a consequence, little is known about the diversification dynamics and morphospace occupation of the Triassic eosauropterygian clades, as well as the existence of phenotypical convergence amongst them. Recent analyses of the temporal trends of vertebrate diversity have highlighted the importance of analyzing regional dynamics, as the structure of the fossil record (*i.e.*, which niches are sampled and how) fluctuates geographically (*Close et al., 2020*; *MacLaren et al., 2022*). Qualitative evidence suggests that Middle Triassic eosauropterygians display geographical differences in their assemblages: pachypleurosauroids found in the Anisian of China (Luoping and Panxian biotas) appear

to have greater morphological diversity, especially in the craniodental region (*Wu et al., 2011*; *Cheng et al., 2012*, *2016*; *Xu et al., 2022*, *2023*) while some European nothosauroids seemed to have developed unique feeding strategies (*Rieppel, 1994*; *de Miguel Chaves, Ortega & Pérez-García, 2018*). In this article, we investigate the cranial and postcranial morphological diversification of Middle Triassic eosauropterygians and explore their patterns of morphospace occupation and possible evolutionary convergence. We also characterize the spatiotemporal distribution of their disparity along the Tethys Ocean.

## MATERIALS AND METHODS

### Data

We gathered a set of thirty-five cranial and postcranial linear measurements (Fig. 1) on thirty-six Triassic eosauropterygian species (17 pachypleurosauroids, 16 nothosauroids and three pistosauroids; see Table S1). We collected data directly from specimens (by a digital calliper with a precision of 0.01 mm), on high-precisions 3D models using Meshlab v2022.02 (*Cignoni et al., 2008*), or using ImageJ (v.1.53) on first-hand pictures and pictures from the literature, when no other alternative was found. The 3D models were generated with a Creaform HandySCAN 300 laser scanner at resolution varying from 0.2 to 0.5 mm, depending on the size of the specimen and with an Artec Eva white light scanner at resolution ≈0.5 mm. These 3D models are available on MorphoSource: https://www.morphosource.org/projects/000508432?locale=en. These measurements were used to calculate twenty-seven-dimension quantitative morphofunctional ratios with clear biomechanical and architectural implications (*Anderson et al., 2011*; *Stubbs & Benton, 2016*; *MacLaren et al., 2017*, *2022*; *Fischer et al., 2020*; *Bennion et al., 2022*). In addition to these ratios, we also added the absolute height of the dental crown, as it represents an informative ecological signal in marine predators (*Fischer et al., 2022*). Finally, we used four discrete traits adapted from *Stubbs & Benton (2016)* to better characterize the morphology of the teeth and the mandible. Twenty-one traits are devoted to craniodental anatomy and 10 sample the postcranial region (see Supplemental Information for the definition and the percentage of completeness of each trait). All species have been submitted to a 40% completeness threshold to prevent any distortions in our ordination analyses caused by an excessive amount of missing data. The counterpart of a such threshold is, however, the exclusion of seemingly peculiar phenotypes (notably *Corosaurus*, *Cymatosaurus*, *Bobosaurus*, and *Paludidraco*). The initial total amount of missing entries in our dataset before applying the threshold equals 21.01%, with respectively 14.81% and 36.11% for the craniodental and postcranial regions.

### Phylogenetic analyses

We generated phylogenetic trees by reanalysing the recently published dataset of *Xu et al. (2022)*, containing 149 characters coded across 50 taxa within maximum parsimony framework, in TNT (v1.5) (*Goloboff & Catalano, 2016*). In order to minimize the impact of homoplasy, we used the implied weighting method to reduce the weight of each character proportionally to their homoplasy. This method seems to be the most adequate in a maximum parsimony framework as it provides more accurate results than equal weighting

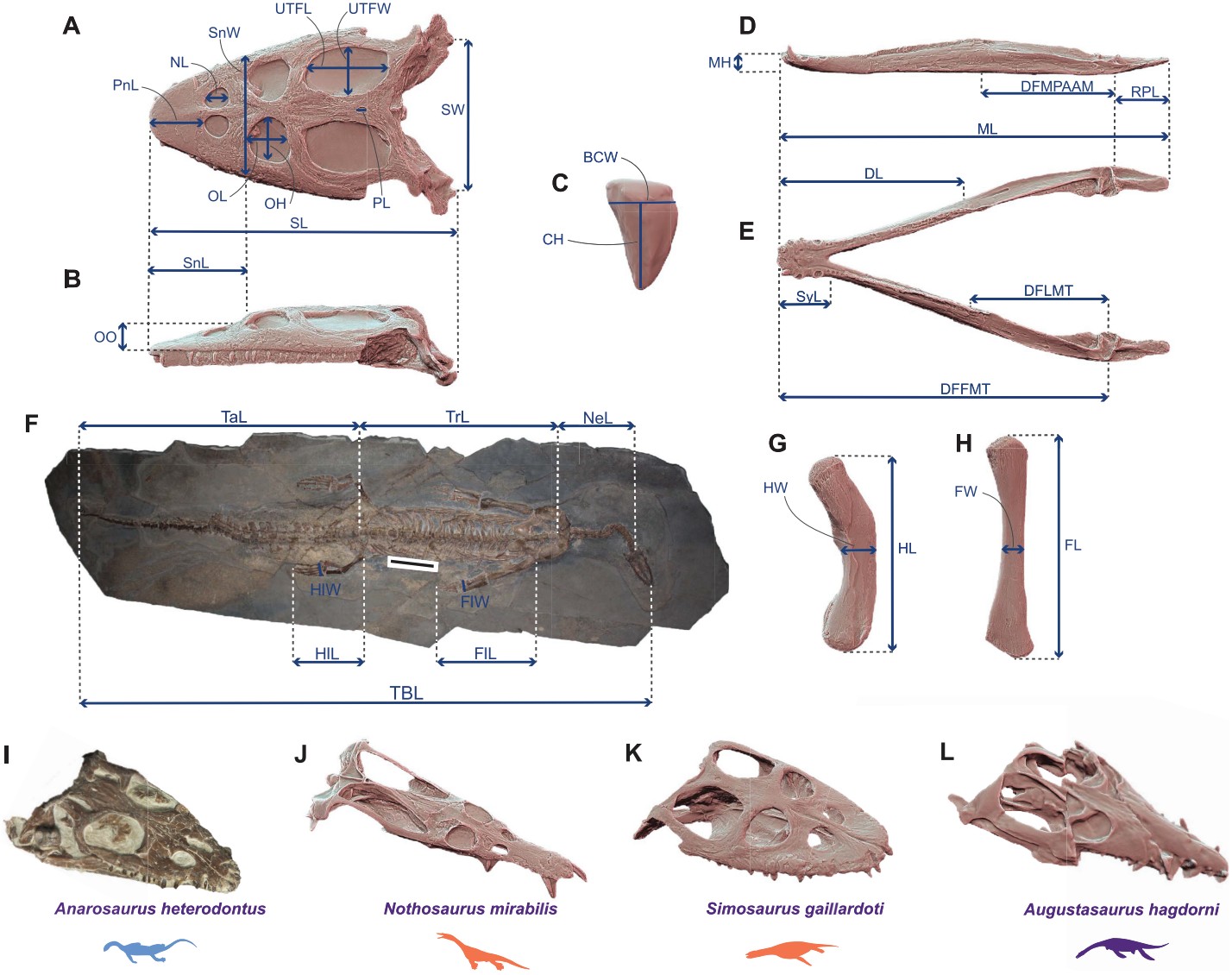

**Figure 1 Linear measurements used to calculate ecomorphological traits and example of Middle Triassic eosauropterygian craniodental architectures.** (A–H) Linear measurements used to compute the ecomorphological traits used in our disparity analyses: (A and B) cranial measurements shown on the 3D model of *Simosaurus gaillardoti* (SMNS 16363) in (A) dorsal and in (B) lateral views; (C) tooth measurements shown on the 3D tooth model of *Simosaurus gaillardoti* (GPIT-PV-60638) in labial view; (D and E) mandibular measurements shown on the 3D model of *Nothosaurus giganteus* (SMNS 18058) in (D) dorsal and in (E) lateral views; (F) postcranial measurements (excepted on humerus and femur) shown on the complete specimen of *Neusticosaurus edwardsii* (PIMUZ T2810); (G) humerus measurements shown on the 3D model of *Nothosaurus giganteus* (SMNS 81311); (H) femur measurements shown on the 3D model of *Nothosaurus giganteus* (SMNS 1589b). (I–L) Examples of Middle Triassic eosauropterygian cranial architectures (I) *Anarodontus heterodontus* (NMNHL RGM443855); (J) *Nothosaurus mirabilis* (SMNS 13155); (K) *Simosaurus gaillardoti* (GPIT-PV-60638); (L) *Augustasaurus hagdorni* (FMNH PR1974). Colors indicate eosauropterygian clade; blue for Pachypleurosauroidea, orange for Nothosauroidea and purple for Pistosauroidea. Abbreviations: BCW, basal crown width; CH; crown height; DFFMT, distance fulcrum—first mandible tooth; DFMLT, distance fulcrum—last mandible tooth; DFMPAAM, distance fulcrum—mid-point of attachment of the adductor muscles; DL, dentigerous length; FL, femur proximodistal length; FW, femur width; FlL, forelimb length; FlW, forelimb width; HL, humerus proximodistal length; HW, humerus width; HlL, hindlimb length; HlW, hindlimb width; MH, mandible height; ML, mandible length; NeL; neck length; NL, naris length; OL, orbit length; OH, orbit height; OO, ocular offset; PL, parietal foramen length; RPL, retroarticular process length; PnL, prenarial length; SL, skull length; SW, skull width; SnL, snout length; SnW, snout width; SyL, symphysial length; TBL, total body length; TaL, tail length; TrL, trunk length; UTFL, upper temporal fenestra length; UTFW, upper temporal fenestra width.

(*Goloboff, Torres & Arias, 2018*; *Smith, 2019*). We decided to use different values of the concavity constant *k* (6, 9 and 12) to test the influence of different character weighting; increasing the *k* value reduces the penalty applied to homoplastic characters which thus play a greater role in estimating phylogenetic relationships.

We set the maximum number of trees to 100,000 and we used the New Technology Search (ratchet activated: 200 iterations; drift activated: 10 cycles; 10 hits and 10 trees per replication). We applied a tree bisection-reconnection (TBR) algorithm on trees recovered by the ratchet to fully explore islands of most parsimonious trees. Our most parsimonious tree, generated with a *k* value of 12, has a length of 25.520 and can be visualized in the Fig. S1. As the phylogenetic dataset of *Xu et al. (2022)* does not include all the species we sampled in our ecomorphological dataset, we added manually six species using the literature and the phytools (v0.7-80) and paleotree (v3.3.25) packages (*Bapst, 2012*; *Revell, 2012*): we split the OTU "*Neusticosaurus*" of the dataset of *Xu et al. (2022)* into its three species, *Neusticosaurus pusillus* as the sister taxa of the clade composed of *Neusticosaurus edwardsii* and *Neusticosaurus peyeri* (*Klein et al., 2022*); *Prosantosaurus scheffoldi* as the sister lineage of the clade comprising *Serpianosaurus* and *Neusticosaurus* (*Klein et al., 2022*); *Brevicaudosaurus jiyangshanensis* as the sister lineage of Nothosauridae (*Shang, Wu & Li, 2020*); *Nothosaurus luopingensis* as the sister lineage of *Nothosaurus yangjuanensis* (*Shang, Li & Wang, 2022*), and *Luopingosaurus imparilis* as the sister lineage of *Honghesaurus longicaudalis* (*Xu et al., 2023*). We pruned the resulting tree by removing all the taxa which have not been included in our ecomorphological dataset, using the ape v5.2 package (*Paradis, Claude & Strimmer, 2004*). The final taxon sampling set can be visualized in the Fig. S2. We then time-scaled it using the minimum branch length algorithm, using a minimal value of 0.5 Myr, using the paleotree package (v3.3.25) (*Bapst, 2012*) (see Fig. S2). The age range of each species of our dataset is provided in Table S1.

## Ordination methods, phylo-ecomorphospace occupation and disparity

All analyses were performed in the R statistical environment (v. 4.2.1) (*R Core Team, 2021*) and followed the protocol established by *Fischer et al. (2020)* which is designed to visualize the density of trait space occupancy and test for the existence of a macroevolutionary landscape. Each continuous trait in the morphological dataset was z-transformed (assigning to all continuous traits a mean of 0 and a variance of 1) prior to computation of a Gower distance matrix. We chose a Gower distance metric as our dataset contains both continuous and discrete traits (*Gower, 1971*). We submitted our distance matrix to a cluster dendrogram analysis using the Ward clustering criterion to visualize the morphological similarities among Triassic eosauropterygians. To evaluate the statistical support of our clustering results, we applied a multiscaled bootstrapping procedure, the 'Approximatively Unbiased *P*-value' method implemented in the pvclust package (v2.2-0) (*Suzuki, Terada & Shimodaira, 2019*). This method creates subsamples of different sizes from our original distance matrix. We ran it from 0.5 to 10 times the size of our distance matrix, at increments of 0.5 and 1,000 bootstraps per increment. We also created tanglegrams (Fig. S5) using the dendextend package (v.1.16.0) (*Galili, 2015*) to compare the phylogenetic position and the phenotypic distance of taxa and we tested their

correlation by computing Mantell tests (1,000 permutations) using the vegan package (v2.5-2) (*Oksanen et al., 2019*). We ran multivariate morphospace analyses *via* both principal coordinate analysis (PCoA) applying the Caillez correction for negative eigenvalues, using the ape package (v5.2) (*Paradis, Claude & Strimmer, 2004*) and non-metric multidimensional scaling (nMDS, dimension = 2), using the vegan package (v2.5-2) (*Oksanen et al., 2019*). We computed phylomorphospaces to visualize the ecomorphological trajectories across the evolution of Triassic eosauropterygians. Density of morphospace occupation was computed using a Kernel two-dimensional density estimate on the PCoA phylomorphospace, using the modified ggphylomorphospace function provided in *Fischer et al. (2020)*. We also displayed the PCoA morphospace occupation in both eastern and western Tethyan realm extracted from the main analyses comprising all taxa for the following time bins of Middle Triassic: Bithynian, Pelsonian, Illyrian (substages of the Anisian) and Fassanian and Longobardian (substages of the Ladinian) for the western and eastern Tethys provinces. The density generated in the main PCoA analysis has also been displayed on these plots. The distributions of skull lengths and widths (the maximum distance between left and right quadrates) are reported in Figs. 2C and 2D, respectively.

## Convergence analyses

We firstly tested the significance of interclade ecomorphological convergence by applying the convergence metrics Ct1, Ct2, Ct3, and Ct4 (*Grossnickle et al., 2023*), which derive from the commonly used metrics of *Stayton (2015)* on selected pairs of taxa based on the results of our ordination analyses. The first two Stayton metrics quantify the phenotypic distance of a pair of taxa by comparison to the dissimilarity of their respective ancestral nodes while the metric C3 and C4 include the total amount of evolution (sum of all phenotypic distances) in the clade defined by the last common ancestor of this pair of taxa. We selected our most parsimonious tree (Fig. S2) to test the significance of these supposed convergences by evaluating the character evolution under 1,000 Brownian simulations using respectively the first two and all axes of the PCoA, generated with the whole-body data. These analyses have been generated using the convevol package (V2.0.0) (*Brightly & Stayton, 2023*). We also used the method developed by *Castiglione et al. (2019)*, using the RRphylo package (2.7.0) (*Castiglione et al., 2018*). This latter is based on whether or not the phenotypic dissimilarity between species tested for convergence (and measurement *via* the angle Ø between their phenotypic vectors) is smaller than expected by their phylogenetic distance under a Brownian Motion model of evolution. In this method, the time spent since cladogenetic divergence represents a crucial factor. We also applied this method using the first two and all axes of the PCoA as for the computation of the Ct metrics.

We decided to test possible ecomorphological convergence between the pistosauroid *Wangosaurus brevirostris* and two nothosauroids which are the closest taxa to this taxon in the dendrogram (Fig. 2A), *Lariosaurus calcagnii* and *Brevicaudosaurus jiyangshanensis*. Even if *Wangosaurus* is phylogenetically found to be the most basal pistosauroid in many analyses (*Ma et al., 2015*; *Jiang et al., 2019*; *Lin et al., 2021*; *Xu et al., 2022, 2023*), its

craniodental architecture and limbs seem quite similar to that of nothosauroids (*Ma et al., 2015*). In our dendrogram (Fig. 2A), the singular nothosauroid *Simosaurus gaillardoti* is found to be morphologically closer to pachypleurosauroids. Therefore, we also decided to investigate the existence of any convergence of *Simosaurus* with the closest pachypleurosauroid in our morphospace, *Qianxisaurus chajiangensis* which also possess a peculiar tooth morphology potentially reflecting a hard-shell prey preference (*Cheng et al., 2012*; *Benton et al., 2013*; *Stubbs & Benton, 2016*). As less than 40% of postcranial information is available for *Simosaurus* (see Table S2), we only use the first two and all axes of the PCoA generated only with all craniodental data (Fig. S10).

## Morphofunctional disparity analyses

We used all axes of PCoA to compute a bootstrapped distribution of the total morphofunctional disparity (sum of ranges, 1,000 Bootstrap iterations) using the dispRity package (v1.2.3) (*Guillerme, 2018*), for both Pachypleurosauroidea and Nothosauroidea in the western and eastern Tethys regions. The significance of difference between the regional disparities for both clades have been calculated with the non-parametric Wilcoxon test. We also calculated the overall morphofunctional disparity for both clades (Pachypleurosauroidea and Nothosauroidea) independently of the location of the taxa (Fig. S14) and for both regions (western and eastern Tethys) without distinguishing the clades (Fig. S15). Given the small number of pistosauroids in our dataset, we decided to not include them in per-clade analyses, but they are sampled for regional disparities (Fig. S15).

# RESULTS

## Cluster dendrogram, morphospace occupation and evolutionary convergence

A clear division in the cluster dendrogram separates species of the dataset into two extremely robust groups (Fig. 2A). The first one comprises all pistosauroids and all nothosauroids (except for *Simosaurus gaillardoti*). In this section of the dendrogram, the primitive pistosauroid *Wangosaurus brevirostris* clusters with two nothosauroids, *Brevicaudosaurus jiyanshanensis* and *Lariosaurus calcagnii*. The second main group in the dendrogram includes all pachypleurosauroids which form a well-defined cluster and *S. gaillardoti*, occupying the most basal position. In the phylo-ecomorphospace (Fig. 2B), the Triassic eosauropterygians tend to globally occupy distinct regions, with the pistosauroids located closer to nothosauroids than to pachypleurosauroids, thus reflecting broad-scale phylogenetic relationships as evidenced by the significant correlation between phenotypic and phylogenetic distances found by our Mantel test (r = 0.687 and *p*-value = 0.001). The separation in the morphospace is mainly due to craniodental morphology; the postcranial skeleton appears less plastic and is marked by a large overlap between pachypleurosauroids and nothosauroids, suggesting a different signal (Figs. S12 and S13). The density of phenotypes recovers two main regions of occupation; one is located at the negative values along the PCoA axis 1 and represents the pachypleurosauroidean morphospace occupation while the other comprises all pistosauroideans and all nothosauroideans with the exception of *Simosaurus*. These two

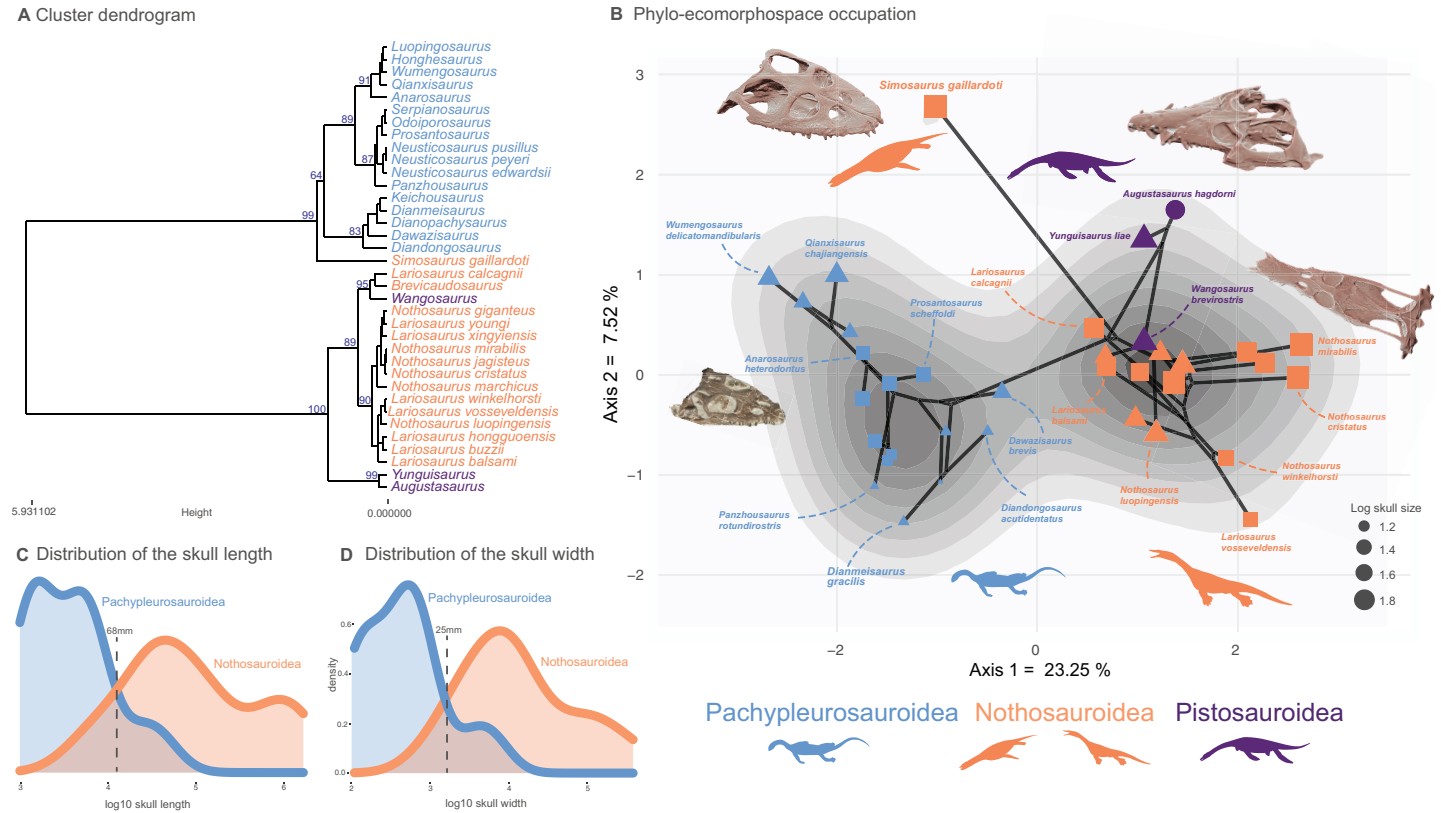

**Figure 2** **Cluster dendrogram, functionnal phylo-ecomorphospace occupation and size distribution of Middle Triassic eosauropterygians.** (A) Cluster dendrogram using the whole-body dataset. Values of the support of the main nodes (approximate unbiased *p*-value in percentage) have been indicated at their corresponding nodes. (B) Phylo-ecomorphospace occupation based on the first two axes of our PCoA analysis using whole-body dataset, superimposed on the density of taxa. Data point sizes scaled to the relative skull size (log skull length). (C and D) Size distribution among pachypleurosauroids and nothosauroids: (C) log skull length, (D) log skull width.

peaks are well separated by a trough with no clear 'intermediate' species sampled in our dataset.

Pachypleurosauroids tend to occupy a wider portion of the morphospace than nothosauroids located in the right peak of density (without *S. gaillardoti*), reflecting a higher degree of morphological variation (see also Fig. S14B). However, the inclusion of the peculiar *Simosaurus* greatly increases the disparity of nothosauroids (Fig. S14), as it occupies a unique region of the ecomorphospace. Indeed, *S. gaillardoti* is characterized by a brevirostrine skull with no rostral constriction, the presence of homodont durophagous dentition, and a relatively small upper temporal fenestra (Fig. 1K) (*Rieppel, 1994*), contrasting with the usually gracile, skulls of nothosaurians characterized by extremely elongated temporal region and specialized heterodont piercing dentition (Fig. 1J) (*Rieppel, 2002*).

As previously mentioned, the position of *Wangosaurus* in the dendrogram and in the morphospace suggests a greater morphological resemblance with nothosauroids than with the more derived pistosauroids such as *Augustasaurus hagdorni* and *Yunguisaurus liae*. Our statistical convergence tests using Ct metrics recover *Wangosaurus brevirostris* as convergent with *Brevicaudosaurus jiyangshanensis* and with *Lariosaurus calcagnii*, its
**Table 1** Results of the convergence tests using the Ct measures derived from the Stayton C-measures for selected pairs of taxa using the first two and all the axes of PCOA analyses on the whole-body dataset and on the craniodental dataset for *Simosaurus gaillardoti* and *Qi. chajiangensis*.

| Taxon pair | PCo axes | Ct1 | *p*-value | Ct2 | *p*-value | Ct3 | *p*-value | Ct4 | *p*-value |
|---|---|---|---|---|---|---|---|---|---|
| *Simosaurus gaillardoti—Qianxisaurus chajiangensis* | PCo 1–2 | 0.108 | 0.260 | 0.030 | 0.215 | 0.043 | 0.269 | 0.010 | 0.171 |
| | All axes | −0.093 | 0.327 | −0.033 | 0.381 | −0.035 | 0.300 | −0.006 | 0.354 |
| *Wangosaurus brevirostris—Lariosaurus calcagnii* | PCo 1–2 | 0.688 | 0.024 | 0.114 | 0.035 | 0.218 | 0.039 | 0.077 | 0.033 |
| | All axes | 0.106 | 0.041 | 0.028 | 0.049 | 0.035 | 0.053 | 0.009 | 0.057 |
| *Wangosaurus brevirostris—Brevicaudosaurus jiyangshanensis* | PCo 1–2 | 0.704 | 0.013 | 0.108 | 0.017 | 0.308 | 0.006 | 0.073 | 0.013 |
| | All axes | 0.106 | 0.013 | 0.026 | 0.013 | 0.040 | 0.013 | 0.008 | 0.021 |
| *Wangosaurus brevirostris* as a basal nothosauroidean—*Lariosaurus calcagnii* | PCo 1–2 | 0.565 | 0.031 | 0.067 | 0.031 | 0.185 | 0.026 | 0.052 | 0.025 |
| | All axes | −0.247 | 0.922 | −0.047 | 0.724 | −0.079 | 0.699 | −0.016 | 0.532 |
| *Wangosaurus brevirostris* as a basal nothosauroidean—*Brevicaudosaurus jiyangshanensis* | PCo 1–2 | −0.229 | 0.025 | −0.008 | 0.025 | −0.066 | 0.035 | −0.007 | 0.024 |
| | All axes | −3.098 | 0.182 | −0.165 | 0.121 | −0.496 | 0.268 | −0.057 | 0.025 |

**Table 2** Results convergence tests by using the Castiglione et al method for selected pairs of taxa using the first two and all the axes of PCoA analyses on the whole-body dataset and on the craniodental dataset for *S. gaillardoti* and *Q.chajiangensis*.

| Taxon pair | PCo axes | Ang.state | Ang.state.time | *p*.ang.state | *p*.ang.state.time |
|---|---|---|---|---|---|
| *Simosaurus gaillardoti—Qianxisaurus chajiangensis* | PCo 1–2 | 46.778 | 0.410 | 0.382 | 0.215 |
| | All axes | 75.96 | 0.666 | 0.377 | 0.044 |
| *Wangosaurus brevirostris—Lariosaurus calcagnii* | PCo 1–2 | 17.768 | 0.655 | 0.174 | 0.082 |
| | All axes | 85.296 | 3.145 | 0.458 | 0.118 |
| *Wangosaurus brevirostris—Brevicaudosaurus jiyangshanensis* | PCo 1–2 | 3.547 | 0.120 | 0.049 | 0.011 |
| | All axes | 72.235 | 2.447 | 0.303 | 0.022 |
| *Wangosaurus brevirostris* as a basal nothosauroidean—*Lariosaurus calcagnii* | PCo 1–2 | 17.768 | 0.211 | 0.174 | 0.092 |
| | All axes | 85.296 | 1.014 | 0.459 | 0.149 |
| *Wangosaurus brevirostris* as a basal nothosauroidean—*Brevicaudosaurus jiyangshanensis* | PCo 1–2 | 3.545 | 0.041 | 0.04 | 0.017 |
| | All axes | 72.235 | 0.835 | 0.310 | 0.036 |

closest relatives in the dendrogram, no matter the number of PCoA axes used (Table 1). The method developed by *Castiglione et al. (2019)* also unambiguously identify phenotypic convergence between *Wangosaurus* and *Brevicaudosaurus* but not with *L. calcagnii* in this case (Table 2). Nevertheless, the significance of phenotypic convergence between *Wangosaurus* and some nothosauroids could be debated due to persisting uncertainties concerning the phylogenetic affinities of *Wangosaurus*. Indeed, this taxon is often recovered as the most basal pistosauroid (*Ma et al., 2015*; *Jiang et al., 2019*; *Lin et al., 2021*; *Xu et al., 2022*, *2023*, as well as our study) but some studies considered it as a basal nothosauroidean instead (*Shang, Wu & Li, 2020*; *Wang et al., 2022*). For this reason, we also tested the morphological convergence of *Wangosaurus* with the two previous taxa by forcing *Wangosaurus* as a nothosauroidean (see Material and Methods, section phylogenetic analyses). The morphological convergence with *Brevicaudosaurus* and *L. calcagnii* are barely noticeable and only recovered by the Ct metrics computed with the

first two axes of the PCoA (Table 1). Furthermore, the method of *Castiglione et al. (2019)* only recovered a significant result with *Brevicaudosaurus* no matter the number of axes used (Tables 2). The combined results of convergence tests in this scenario suggest that the morphological similarity between *Wangosaurus* and *Brevicaudosaurus* would be more explained by conservatism rather than by a truly convergence. Evidence of craniodental convergence between *Simosaurus gaillardoti* and *Qianxisaurus chajiangensis* is almost absent and is only recovered with the method of *Castiglione et al. (2019)* when using all axes of the PCoA (Tables 1 and 2). The results of our convergent analyses thus only unambiguously highlight a phenotypical convergence between *Brevicaudosaurus* and *Wangosaurus* when considered as a pistosauroid whereas other case of convergence tested in this paper should be interpreted cautiously.

### Regional and temporal patterns of disparity

Pachypleurosauroids and nothosauroids each evolved an approximate equal amount of disparity, even if nothosauroids appears to be slightly more disparate ($p$-value < 0.001) (Fig. S14A). This difference in magnitude is mainly due to the unique craniodental morphology of *S. gaillardoti*. By removing this taxon and comparing pachypleurosauroideans and nothosauroideans present in the high-density area located on the positive values along the PCoA axis 1 (Fig. 2B), pachypleurosauroids appear much more diverse ecomorphologically ($p$-value < 0.001) (Fig. S14B). The western Tethyan faunal province records a greater amount of disparity than the eastern Tethyan one ($p$-value < 0.001) (Fig. S15A), but this difference is once again exaggerated by the presence of *Simosaurus* ($p$-value < 0.001) (Fig. S15B). Our results also show a strong geographical differentiation in the amount of ecomorphological disparity of the two clades. Pachypleurosauroids are clearly more disparate in the eastern Tethyan realm ($p$-value < 0.001) (Fig. 3A) whereas nothosauroids have a disparity maximum in the western Tethyan realm ($p$-value = 0) (Fig. 3B), even with the absence of *Simosaurus* ($p$-value = 0) (Fig. S16). These regional patterns can be visualized in the morphospace occupation of each geographical regions (Figs. 3D and 3J). Furthermore, the temporal evolution of disparity seems to also vary within these two regions (Figs. 3D–3L). In the western Tethyan realm, the greatest eosauropterygian ecomorphological diversification occurs during the Fassanian (early Ladinian; Fig. 3H), while the maximum of disparity in the eastern Tethyan realm is recorded during the Pelsonian (middle Anisian), with the diversification of pachypleurosauroids (Fig. 3K). All these results thus tend to highlight a strong geographical and temporal decoupling in the ecomorphological diversification of Triassic eosauropterygians.

## DISCUSSION

### The early evolutionary trajectories of Triassic eosauropterygians reflect dietary specialization

Our ordination and convergences analyses provide new insights on the diversification of Triassic eosauropterygians and reveal a global colonization of distinct ecomorphospaces for each clade (with the exception of *Wangosaurus*). This lack of overlap during their

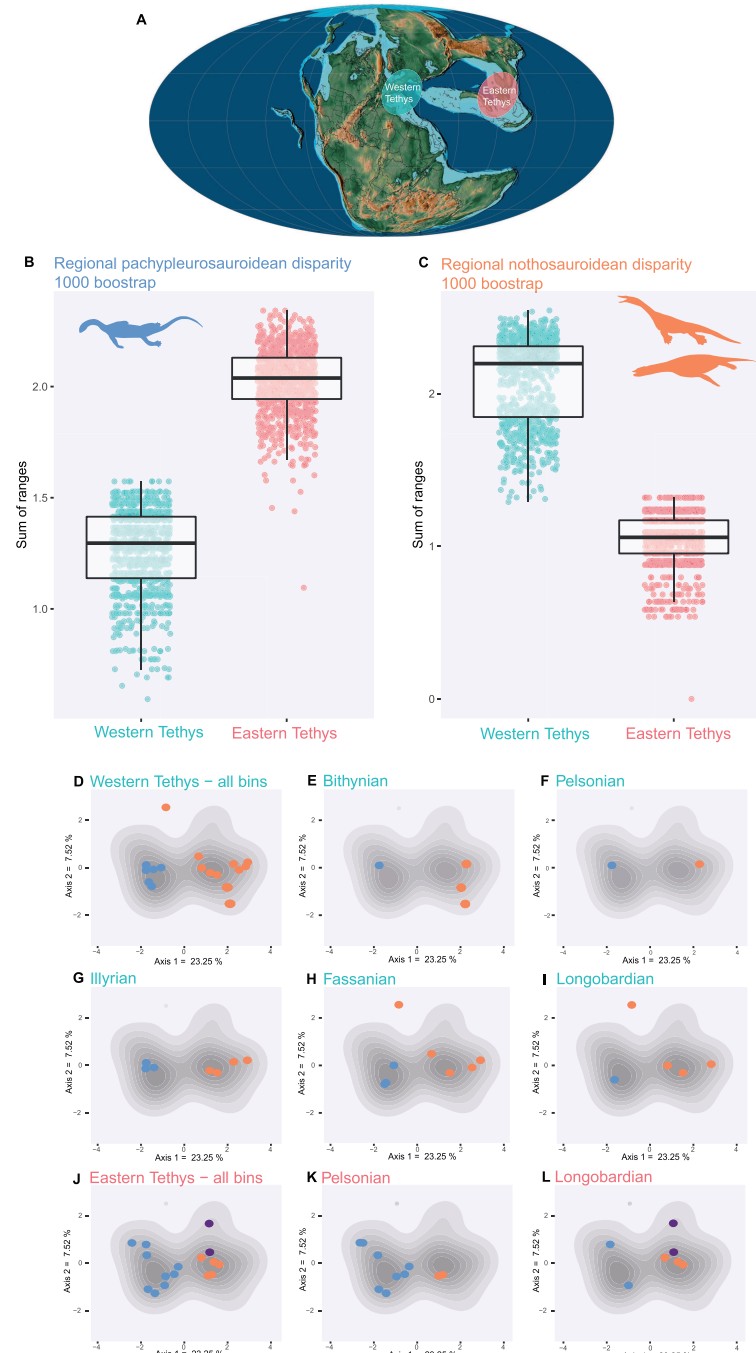

**Figure 3 Regional and temporal pachypleurosauroids and nothosauroids pattern of ecomorphological disparity.** (A) Paleobiogeography of the Middle Triassic (Ladinian) modified from *Scotese (2016)*. (B and C) Comparison of the total disparity between the western and eastern Tethyan realm (B) for pachypleurosauroids and (C) nothosauroids. (D–L) Eosauropterygian ecomorphospace occupation through time during the Middle Triassic superimposed on the density of taxa (grey shades). (D–I) In the western Tethys and (J–L) in the eastern Tethys. Bithynian, Pelsonian and Illyrian are time bins of the Anisian while Fassanian and Longobardian are time bins of the Ladinian. Eastern Tethys eosauropterygians have only been found in the Pelsonian and Longobardian, in the Luoping, Panxian and Xingyi biotas respectively.

"early burst" radiation (*Stubbs & Benton, 2016*) reflects the influence of developmental constraints among clades and supports the inference of a substantial and rapid trophic specialization in eosauropterygians, mainly reflected in their craniodental architecture and body size (*Rieppel, 2002*). In a previous study which aimed to investigate the feeding mechanics in Triassic early sauropterygians by reconstructing their jaw adductor musculature, *Rieppel (2002)* highlighted the near absence of an overlap in the inferred feeding strategies of Triassic eosauropterygians. On one hand, the craniodental architecture of pachypleurosauroids, which are usually small-sized (rarely exceeding 50 cm according *Rieppel (2000))* is indeed characterized by a rounded, short and blunt snout, a very short symphysis and a homodont peg-like dentition, strongly suggesting that they have captured their prey by suction feeding, followed by a rapid snapping bite (*Rieppel, 2002*; *Xu et al., 2023*). Pachypleurosauroids likely evolved a wider array of feeding strategies through time, however. *Xu et al. (2023)* reported a progressive reduction in the length of the hyoid apparatus and an increase in snout length, involving a gradual shift away from suction feeding for some derived species mostly in eastern Tethys (*Wu et al., 2011*; *Cheng et al., 2012*; *Xu et al., 2023*). On the other hand, nothosauroids and pistosauroids likely used their narrow and 'pincer jaw', to conduct sideward-directed snapping bites (*Rieppel, 2002*). Their skull morphology and dentition suggest, here also, a range of food procurement strategies (*Rieppel, 2002*). The dentition and the cranial architecture of pistosauroids should be more suitable to puncture prey (*Rieppel, 2002*), while the presence of procumbent and enlarged fangs in nothosauroids might have served as a fish-trap (*Chatterjee & Small, 1989*; *Rieppel, 2002*). Large species such as *L. calcagnii* and *N. giganteus* or *N. zhangi* could have nevertheless occupied the top of the food chain in their ecosystems and preyed on smaller marine reptiles (*Tschanz, 1989*; *Rieppel, 2002*; *Liu et al., 2014*). The isolated position of *S. gaillardoti* with respect to other nothosauroids in the ecomorphospace is mainly explained by its unique feeding strategy as it is the only eosauropterygian to have developed a durophagous dentition to crush hard-shelled preys such as ammonoids or hard-scaled fishes (*Rieppel, 1994*, *2002*; *Klein & Griebeler, 2016*; *Klein, Eggmaier & Hagdorn, 2022*). The magnitude of durophagy in *Simosaurus* is however not comparable to that of placodont sauropterygians which possess a much more robust mandible and teeth highly modified into low labial bulbs and lingual tooth plates (*Rieppel, 2002*; *Neenan, Klein & Scheyer, 2013*; *Neenan et al., 2015*).

The postcranial anatomy of Triassic eosauropterygians appears to be less plastic than their craniodental skeleton, which is also possibly the case in short-necked plesiosaurians (besides relative neck length) (*O'Keefe, 2002*; *Fischer et al., 2020*). This homogeneity in the morphological diversification of the postcranial region may reflect a conservative locomotion mode in shallow water intraplatform basins, through full body oscillation (*Carroll, Gaskill & Whittington, 1985*; *Rieppel & Lin, 1995*; *Rieppel, 2000*; *Neenan et al., 2017*; *Krahl, 2021*; *Xu et al., 2022*; *Klein, Eggmaier & Hagdorn, 2022*). However, recent studies suggest the use of forelimbs in propulsion among nothosauroids (*Zhang et al., 2014*; *Klein et al., 2015*; *Krahl, 2021*; *Klein, Eggmaier & Hagdorn, 2022*), thus contrasting with the strict anguilliform swimming seen in pachypleurosauroids (*Sander, 1989*; *Xu et al., 2022*; *Gutarra & Rahman, 2022*). It is noteworthy to mention that pistosauroids

occupied a portion of the ecomorphospace that has not been colonized by any other eosauropterygians, possibly reflecting the transition from the undulating non-pistosauroids to the pelagic paraxial swimming seen in plesiosaurians (*Sato et al., 2014b*).

Many of the species we analysed coexisted, suggesting that the range of morphologies occupied reflects colonization of multiple niches and, perhaps, niche partitioning. The fairly rapid ecomorphological specialization of eosauropterygians is probably best understood in the context of increasing complexity of marine trophic webs of the shallow marine and coastal environments of the Tethys during the Middle Triassic, following the recovery of the PTME (*Benton et al., 2013*; *Scheyer et al., 2014*; *Liu et al., 2014*, *2021*; *Li & Liu, 2020*). Indeed, such a diversification pattern is not exclusive of eosauropterygians and have also been detected in ichthyosaurians, tanystropheids, and saurichthyid fishes as well (*Benton et al., 2013*; *Wu, Sun & Fang, 2017*; *Spiekman et al., 2020*; *Bindellini et al., 2021*).

## Regional diversification patterns in eosauropterygians

Our results quantitatively reveal a pervasive regional difference in the disparity of the eosauropterygian assemblages along the Tethys margins suggesting a different biogeographical diversification for pachypleurosauroids and nothosauroids. Pachypleurosauroids seemed to undergo a remarkable ecomorphological radiation during the Pelsonian in the eastern Tethyan realm, soon after their earliest appearance in the fossil record of that region (*Jiang et al., 2014*). This diversification mostly occurred in the Luoping, but also in the Panxian biotas, leading to the coexistence of numerous species with distinct feeding strategies (see the craniodental architecture of *Luopingosaurus*, *Diandongosaurus* or *Wumengosaurus*) or unique swimming capabilities among pachypleurosauroids (*e.g.*, *Honghesaurus*) (*Wu et al., 2011*; *Benton et al., 2013*; *Sato et al., 2014a*; *Shang & Li, 2015*; *Cheng et al., 2016*; *Liu et al., 2021*; *Xu et al., 2022*, *2023*). By comparison, European and more specifically the Alpine pachypleurosauroids are morphologically more similar (*Rieppel, 2000*; *Renesto, Binelli & Hagdorn, 2014*; *Beardmore & Furrer, 2016*; *Klein et al., 2022*) and thus characterized by lower values of disparity. However, the validity of the taxonomic variability of Chinese pachypleurosauroids could be questioned. With the exception of the abundant *Keichousaurus hui* for which the ontogenetic series is well known (*Lin & Rieppel, 1998*; *Cheng et al., 2009*), a series of species are based on a single specimen and are in need of a thorough taxonomic reinvestigation. This possible overestimation of Chinese pachypleurosauroids could also affect regional disparity patterns.

Nothosauroids have been found to be more abundant and disparate in the western Tethys in comparison to their eastern Tethys relatives. The total ecomorphological disparity of European nothosauroids is potentially underestimated in our analyses by the absence of the peculiar but fragmentary simosaurid *Paludidraco multidentatus* from the Upper Triassic of Spain (*de Miguel Chaves, Ortega & Pérez-García, 2018*), whose unique anatomy suggest a manatee-like feeding and locomotion mode (*de Miguel Chaves, Ortega & Pérez-García, 2018*). The unique morphologies of *Paludidraco* and *Simosaurus* would attest to the higher potential of diversification in feeding strategies in primitive European

nothosauroids, compared to the more derived nothosaurians which appeared more similar, excepted in their size (*Rieppel & Wild, 1996*; *Liu et al., 2014*).

Variation in the quality of the fossil record, notably Lagerstätten effects can be a powerful driver of spatiotemporal differences in diversity and can have complex impacts on disparity trends (*Benson et al., 2010*; *Benson & Butler, 2011*; *Sutherland et al., 2019*). In our study, both the eastern and western Tethys localities have been intensively sampled overtime, especially in Luoping, Panxian, Xingyi (China), and Monte San Giorgio and Winterswijk (Europe), allowing comparisons between these two regions (*Rieppel, 2000*; *Furrer, 2003*; *Motani et al., 2008*; *Benton et al., 2013*; *Renesto, Binelli & Hagdorn, 2014*; *Heijne, Klein & Sander, 2019*). Nevertheless, the Chinese assemblages which have been characterized as exceptional in terms of faunal communities represent temporally disconnected 'snapshots' of the Pelsonian (Luoping and Panxian) and Longobardian (Xingyi) while European localities have produced a rather continuous Middle Triassic time series (*Rieppel, 2000*; *Röhl et al., 2001*; *Furrer, 2003*; *Hu et al., 2011*; *Benton et al., 2013*).

Thus, if not spatial heterogeneities of the fossil record, what drives the observed differences among eosauropterygian phenotypes between Tethysian provinces? Both regions were likely subtropical shallow platform environments characterized by a rather similar vertebrate assemblages mainly composed of saurichthyid fishes (*Wu et al., 2009*; *Benton et al., 2013*; *Maxwell et al., 2015*, *2016*), mixosaurid ichthyosaurians (*Brinkmann, 1998*; *Motani, 1999a*; *Maisch, Matzke & Brinkmann, 2006*; *Jiang et al., 2006*; *Benton et al., 2013*; *Liu et al., 2013*), placodonts (*Neenan, Klein & Scheyer, 2013*; *Neenan et al., 2015*), thalattosaurians (*Cheng, 2003*; *Müller, 2005*; *Cheng et al., 2010*), and tanystropheids (*Rieppel, Li & Fraser, 2008*; *Spiekman et al., 2020*), in addition to eosauropterygians. This broad-brush homogeneity in the faunae is expected to result from a dispersal route along coastlines of the Tethys, allowing exchanges between the European and Chinese provinces (*Rieppel, 1999*; *Bardet et al., 2014*).

Drivers of this decoupling in disparity among pachypleurosauroids and nothosauroids remains unclear and require a thorough reinvestigation of the differences between ecosystems along the Tethys Ocean. However, this variation reveals the importance of analysing regional dynamics rather than a summed-up, oversimplified signal when spatial heterogeneities appeared strong, as recently demonstrated by *Close et al. (2020)* and *MacLaren et al. (2022)*.

## Eosauropterygia, a plastic clade throughout most of its history

Eosauropterygia and Ichthyosauria were the longest-lived clades of Mesozoic marine reptiles. The shape of their radiations and subsequent diversifications has been analysed in terms of skull size, mandible shape, and skeletal characters suggesting an early-burst radiation that produced a variety of morphologies in the shallow marine environments during the Middle Triassic (*Stubbs & Benton, 2016*; *Moon & Stubbs, 2020*). However, the remodelling of marine ecosystems caused by regression events during the Late Triassic profoundly altered their evolutionary histories, with only pelagic morphotypes surviving across the Triassic–Jurassic boundary (*Benson et al., 2010*; *Benson & Butler, 2011*; *Thorne, Ruta & Benton, 2011*; *Dick & Maxwell, 2015*; *Wintrich et al., 2017*). These selective

extinctions forced a quantitative drop in the disparity but are coupled with the emergence of parvipelvians and plesiosaurians during the late(st) Triassic (*Motani, 1999b*; *Dick & Maxwell, 2015*; *Stubbs & Benton, 2016*; *Wintrich et al., 2017*; *Moon & Stubbs, 2020*). While the disparity of ichthyosaurian surviving lineages seems to be be considerably reduced in comparison to their Triassic ancestors (*Thorne, Ruta & Benton, 2011*; *Dick & Maxwell, 2015*; *Fischer et al., 2016*), post-Early Jurassic plesiosaurians has been characterized by an impressive ecomorphological diversity (*O'Keefe, 2002*; *Benson & Druckenmiller, 2014*; *Fischer et al., 2020*). Indeed, the evolutionary history of plesiosaurians has been marked by the iterative evolution of superficially similar phenotypes (*e.g.*, 'plesiosauromorph' *vs* 'pliosauromorph', 'longirostrine' *vs* 'latirostrine' in short-necked plesiosaurians) over time (*O'Keefe, 2002*; *Fischer et al., 2017*, *2018*, *2020*) and by the ability to innovate in their swimming and feeding strategies in their late evolution (*Robin O'Keefe et al., 2017*).

This great ecomorphological diversification demonstrates that plesiosaurians were continuously capable of producing a large variety of forms and were therefore characterized by a high phenotypic plasticity which may have helped them to withstand or adapt to changes in the ecosystems over the Jurassic and the Cretaceous. The remarkable feeding specialization among Middle Triassic eosauropterygians coupled with their distinct regional patterns of diversification also highlight a such phenotypic plasticity in addition to their high developmental plasticity identified by the diversity of their life history traits (*Klein & Griebeler, 2018*; *Griebeler & Klein, 2019*). Our results would thus suggest that eosauropterygians have always displayed a wide range of craniodental architectures and that a high morphological plasticity has characterized their overall evolutionary history; the initial plesiosaurian radiation during the Early Jurassic being the exception with the lowest values of disparity recorded (*Benson, Evans & Druckenmiller, 2012*; *Benson & Druckenmiller, 2014*; *Stubbs & Benton, 2016*).

## CONCLUSIONS

In this article, we reinvestigate the ecomorphological diversification of Middle Triassic eosauropterygians. We found that this important diversification led to craniodental distinction and feeding specializations among pachypleurosauroids, nothosauroids and pistosauroids, suggesting low interspecific competition in the shallow intraplatform basins bordering the Tethys Ocean. On the other hand, our results indicate that their postcranial anatomy appear more homogeneous, mainly between pachypleurosauroids and nothosauroids. This trend suggests a decoupling in the evolution of these two anatomical regions, similarly to what has been proposed for derived short-necked plesiosaurians. Our analyses also demonstrate that the disparity of pachypleurosauroids and nothosauroids differs along the Tethys margins, reflecting regional variations in their disparity. The eastern Tethys during the Pelsonian represented a unique 'hotspot' for the morphological diversification of pachypleurosauroids in which various craniodentally distinct taxa co-occurred. The western margin of the Tethys was dominated by nothosauroids, and their disparity has been mainly increased by the morphology of *Simosaurus*. This regional variation in disparity would suggests that Triassic eosauropterygians diversified in a different way depending on the biotic and abiotic

features of the ecosystems. This high phenotypic plasticity also characterizes the evolution of post-Triassic plesiosaurians, casting the entire Eosauropterygia as a particularly plastic clade.

## ACKNOWLEDGEMENTS

We would like to thank all the museum curators and staff for granting us access to their specimens. We thank Dr. Ingmar Werneburg and Dr. Anne Krahl (Paleontology Collection in Tübingen, GPIT); Dr. Christian Klug (Paläontologisches Institut der Universität Zürich, PIMUZ), Dr. Erin Maxwell (Staatliches Museum für Naturkunde Stuttgart, SMNS), Natasja den Ouden (National Museum of Natural History Leiden (Naturalis); NMNHL), William Simpson (Field Museum National History, FMNH). We also want to thank Dr. Jamie MacLaren for helping in creating the ecomorphological traits, Narimane Chatar for her help and the discussion about the code R and the two reviewers, Dr. Brenen Wynd and Dr. Carlos de Miguel Chaves for their thorough comments which have greatly improved the quality of the article.

### Funding

This work was supported by the the Fonds de la Recherche Scientifique doctoral (F.R. S–FNRS) FRIA grant (No. FC38761) and the Swiss National Science Foundation (No. 31003A_179401). The funders had no role in study design, data collection and analysis, decision to publish, or preparation of the manuscript.

### Grant Disclosures

The following grant information was disclosed by the authors:
Fond de la Recherche Scientifique doctoral (F.R.S–FNRS) FRIA Grant: FC38761.
Swiss National Science Foundation: 31003A_179401.

### Competing Interests

The authors declare that they have no competing interests.

### Author Contributions

- Antoine Laboury conceived and designed the experiments, performed the experiments, analyzed the data, prepared figures and/or tables, authored or reviewed drafts of the article, and approved the final draft.
- Torsten M. Scheyer conceived and designed the experiments, performed the experiments, authored or reviewed drafts of the article, and approved the final draft.
- Nicole Klein conceived and designed the experiments, performed the experiments, authored or reviewed drafts of the article, and approved the final draft.
- Thomas L. Stubbs conceived and designed the experiments, performed the experiments, analyzed the data, authored or reviewed drafts of the article, and approved the final draft.

- Valentin Fischer conceived and designed the experiments, performed the experiments, analyzed the data, prepared figures and/or tables, authored or reviewed drafts of the article, and approved the final draft.

## Data Availability

A summary of the workflow analyses made in RStudio, raw measurements, temporal and regional data are available in the Supplemental Files.

The 3D models are available at MorphoSource:

- Media 000510964: Right Humerus; DOI 10.17602/M2/M510964.
- Media 000510969: Right Humerus; DOI 10.17602/M2/M510969.
- Media 000510974: Lower Jaw; DOI 10.17602/M2/M510974.
- Media 000510979: Right Humerus; DOI 10.17602/M2/M510979.
- Media 000510985: Cranium; DOI 10.17602/M2/M510985.
- Media 000510990: Cranium; DOI 10.17602/M2/M510990.
- Media 000510995: Cranium; DOI 10.17602/M2/M510995.
- Media 000511000: Right Humerus; DOI 10.17602/M2/M511000.
- Media 000510680: Cranium; DOI 10.17602/M2/M510680.
- Media 000510688: Left Humerus; DOI 10.17602/M2/M510688.
- Media 000510693: Left Humerus; DOI 10.17602/M2/M510693.
- Media 000510711: Right Femur; DOI 10.17602/M2/M510711.
. Media 000510733: Right Humerus; DOI 10.17602/M2/M510733.
- Media 000510747: Femur; DOI 10.17602/M2/M510747.
- Media 000510757: Left Humerus; DOI 10.17602/M2/M510757.
- Media 000510959: Lower Jaw; DOI 10.17602/M2/M510959.
- Media 000510627: Cranium; DOI 10.17602/M2/M510627.
- Media 000510648: Anterior Vertebrae, Cranium, Mandible; DOI 10.17602/M2/M510648.
- Media 000510652: Lower Jaw; DOI 10.17602/M2/M510652.
- Media 000510657: Cranium; DOI 10.17602/M2/M510657.
- Media 000510661: Cranium; DOI 10.17602/M2/M510661.
- Media 000510670: Lower Jaw; DOI 10.17602/M2/M510670.
- Media 000510675: Cranium; DOI 10.17602/M2/M510675.

## Supplemental Information

Supplemental information for this article can be found online at http://dx.doi.org/10.7717/peerj.15776#supplemental-information.

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
