# Peer review of "High phenotypic plasticity at the dawn of the eosauropterygian radiation"

_PeerJ, doi:10.7717/peerj.15776_

## Round 0.1 · original submission · Major Revisions

I have now received back the comments by two reviewers. Most of their comments were minor, but nevertheless suggest many aspects of the writing to be improved. Notice that R1 makes important points regarding data analysis and presentation. I fully agree with him that a PCoA is much better than nMDS to represent the morphospace. The same reviewer also requires that you provide more information as to turn your data analysis workflow more reproducible. Instead of providing a .Rdata file, I'd highly suggest authors to consider depositing their data and an R Markdown/Quarto dynamic document to document your analysis.

I'm not sure if Table 1 and 2 should be kept in the main text given that they're not reporting the results of any analysis, consider including them suppl mat.
Consider re-working the abstract as to focus more on the results and conclusion, instead of introduction.

You should definitely consider re-running convergent evolution analysis with alternative metrics, such as Ct-measures, as suggested by R1, but also theta (https://journals.plos.org/plosone/article?id=10.1371/journal.pone.0226949) in order to test if the results are robust independent of the metric used.
Avoid long citation strings in the introduction and discussion. Do not cite figures in the discussion.

Cite the ref for Gower coefficient. It seems odd to be calling the phylomorphospace a "macroevolutionary landscape", since this is simply an ordination diagram that summarizes the phenotypic data with the phylogeny superimposed. By macroevolutionary landscape I'd think more of an analysis derived from an OU model. R1 indeed suggests you to fit an OU model using Bayou R package. However, this package is only for univariate data, not multivariate like yours. So, consider using mvMORPH or phylogeneticEM instead, if you're considering using these analysis.

Your Mantel test (L. 154-8) between the phylogeny and the phenotypic distance matrix is what we usually would call a phylogenetic signal analysis. But this is not shown in the results.

No need to say "cluster dendrogram", just refer to it simply as dendrogram.
L. 255-66: always report test statistics along with the respective P value, and degrees of freedom when appropriate.

·

Excellent Review

This review has been rated excellent by staff (in the top 15% of reviews)
EDITOR COMMENT
This was a very useful and detailed review. It pointed out not only the deficiencies, but also highlighted the strengths of the paper. It definitely helped me making my decision.

Basic reporting

The authors present an impressive array of analyses to better estimate the tempo and magnitude of morphological diversification in Eosauropterygia, a clade of secondarily aquatic reptiles that evolve following the Permian-Triassic mass extinction. The authors work largely with 3D data taken from physical specimens, images, and 3-D surface scanning, and I greatly appreciate the authors including a direct hyperlink to the repository that houses their scans. I think these kinds of efforts are extremely important in making science more accessible, and it is great to see these authors making these wonderful data accessible. The authors perform a phylogenetic analysis based on previous analyses. I present my most salient points, in order of importance here, and follow with my comments by line. I think that minor revisions are necessary and this absolutely should be accepted for publication in PeerJ.

1. The authors use a combination of linear and discrete metrics to summarize cranial and postcranial diversity in early-diverging sauropterygians, and subject them to ordination (nMDS and PCoA) to represent their data visually. By in large, I agree with the author's decisions; however, not in the use of nMDS as the basis for an adaptive-landscape-esque ecophylomorphospace. I present my argument below, and I suggest that they use the PCoA ordinations in lieu of the nMDS (which code for these is included in the R file, but as far as I can see, none of the PCoA phylomorphospace plots are presented in this way in the main text or supplement). My justification of this is that the nMDS creates an abstracted form of the data that ranks them based on similarity, so their placement and distance between points is based on this abstraction, and is not based directly on the underlying data. Because the authors have already done these analyses it should be straightforward to provide new figures.

2. I think that the current supplementary data files are presently insufficient for suitable reproducibility, and will need to be corrected prior to publication. Specifically, no nexus file is presented for the phylogenetic analysis, and the supplied R code includes every analysis performed, but does not sufficiently document what the fundamental steps of each analysis are and why they are done. Some comments are provided, and these are appreciated, but by in large, vertebrate paleontologists are not very familiar with R, and would have an incredibly difficult time trying to adapt these analyses to their own works. To fix this, I think the authors should not include a .R file with all of the code run for this paper, but instead with one example for each major analysis that is heavily commented. This would ensure that the code is more easily adapted by new authors, or more easily reviewable.

3. The phylogenetics are good, but I point out a few pieces of information that are missing in the Materials + Methods, which can be important for recreating, building off of, or even summarizing their phylogenetic approach in future projects.

I present the majority of my comments in the additional comments section, designated by line.

I understand that I am criticizing a large portion of the analysis, and so I would like to welcome the authors to reach out to me if that they have questions or concerns regarding my review.

Very great work and I am extremely excited to see the final, published product!

Best wishes,

Brenen M Wynd

(brenen.wynd@selu.edu)

Experimental design

Discussed at length elsewhere in this review.

Validity of the findings

No comment, the authors did a great job.

Additional comments

Comments by line:

118 - please include a line that clearly states the number of taxa and the number of characters in the matrix.

119 - can you provide a justification for why you chose implied weights? Did you also do a standard tree estimation with no weighting and see if the clades more or less come out as similar between analyses?

122 - please give a very brief explanation behind the concavity constant, and that characters that change more (more transitions) are less impactful in tree estimation and given a lower weight when constructing trees. I would also explicity state that the concavity constant sets the threshold for the implied weights and that low values (k=0-2) make homoplasy a very minor player in estimating phylogenies, and a K of 12, would mean that homoplastic characters do carry some weight in the topologies presented.

136 - please list the taxa or refer to a supplementary table where you show the final taxon-sampling set.

166 - This currently reads as you are performing independent nMDS analyses on the taxa from each time bin. It looks like this isn't the case based on the supplement, but changing this wording to reflect that the time bins are essentially just convex hulls extracted from a parent-nMDS plot, would be important. With the current reading, and suggestion of performing separate nMDS ordinations on each time bin, there would be no way to compare those plots to one another in a meaningful way, because they aren't mathematical solutions to the matrix in question.

170 - It may be worthwhile to check how your calculations differ from those recently presented on BioRxiv (https://www.biorxiv.org/content/10.1101/2022.10.18.512739v2.abstract), where CT Stayton and other authors amend these metrics to better account for edge cases.

141-193 - please specify which ordinations you are using for these analyses. I would like to apologize in advance for the length of the following paragraph, and I do hope you read this as my intent to make this work stronger, and not as a criticism. I think this is excellent work, and I believe that my thoughts regarding ordination could be useful in this regard.

I see that this builds off of previous work from the last author, and I point this out because I fully understand the development and later implementation of methods (I have done the same, and so I am wholly sympathetic). I think this work is well done and compelling, but I have a philosophical disagreement in the use of nMDS as an ecomorphospace to be used in generating hypothetical adaptive landscapes or to calculate the convergence analyses. I don't want this to come across as rude, but more so that I believe the use of nMDS in this way can be misleading and can generate somewhat erroneous clusters/results. Furthermore, the iterative nature of the nMDS means that two separate analyses on the exact same data would produce different adaptive zone widths, as the positioning of taxa will be different as long as a unique seed is used to generate each nMDS. This is my chief concern, the distance between points in nMDS has no euclidean basis and is a representation of the clustered dendrogram in two (unless more are specified) dimensions. Because the positioning and distance between taxa is not based directly on the data, but an abstracction of the data that does not have the capability to fully represent the original data. I say this because the PCoA used in these analyses would be perfect substitutions for the nMDS figures. Since the distances between and amongst points are based on maximizing the variance from euclidean distances between all individuals of each feature. I base my reasoning largely off of [1-3], with 3 being a very strong use of similar ideas (though using semi-landmark data instead of linear morphometrics) that have been incorporated into the paleontological literature (based on [4], but [3] is the first published form of this particular method, that I am aware of, in paleo). To reiterate, I strongly caution against the use of nMDS here as a macroevolutionary landscape. The results of an nMDS are stochastic, based on its iterative nature of the ranked order of features in estimating and visually representing similarity, and are dependent on the number of axes the model is permitted to explore. I hope that this provides some clarity on my position and I would be happy to chat further with the authors about this, if they would like to further discuss my stance.

182 - Fig. 2B is called before Fig. 2A is called. Either change the ordering of the figure to make the Dendrogram panel A, or provide a citation to Fig. 2A somewhere before line 182.

190 - I think it would be important to move this statement about using the PCoA to the top of this section, just after the topic sentence, to clarify which data are being analyzed for this section of the manuscript.

226 - as it currently reads, I am unable to readily differentiate between whether or not the text is referring to nMDS or PCoA. I think this needs to be more explicit, as PCoA should be the center of discussion around variance, and nMDS should be primarily discussing similarity.

279 - I do not agree with the statement that "Our ordination analyses provide new quantitative insights..." These ordinations are visual representations of the data and are not inherently analyses. Especially since the only quantitative metric I can see extracted from these would be the Stayton convergence estimations. I think this can be reduced to "Our ordinations and convergence analyses provide new insights on the..."

353 - "...sometimes with only a single specimen, without..." ?

403 - Was it only pelagic vertebrates that survived? This reads as all animals, so it may be good to add the vertebrate clarifier if that is what you are trying to communicate.

799 - Rieppel, Olivier. "Osteology of Simosaurus gaillardoti and the relationships of stem-group Sauropterygia." Fieldiana (Geology), ns 28 (1994): 1-85.

879 - Stayton CT. 2017. Convevol: analysis of convergent evolution. R package version 1:471.
- keep the format of citations the same and make sure you are using the correct citation (I found this one on scholar and it included a bit more information about the package, even though it is minor)

915 - I suggest changing the capitalization of the journal name so that all journal names are represented in the same case (e.g., sentence-case) through the entire references section

Fig 3 - Why do the Bithynian and Pelsonian time periods include a blue dot in the bottom-lefthand corner of the plots (E+F), that is not present in the all bins plot (D)?

Fig 3 - I am having a difficult time with sections D-L of fig 3 being represented as an ecomorphospace. My issue is that the all bins section (D), is homogenizing the proposed ecomorphospace of multiple different time bins and thus removing signal for how the ecomorphospace changes through time. For example, because D includes all species in the analysis, it is likely going to cluster ecologically similar taxa near one another. But what if one species in time-bin 'A' is directly replaced by another organism in time-bin 'B'. In this scenario, the overall landscape of the ecomorphospace may not change, and there may still only be a single narrow peak that that taxon exists on, and is later replaced post-extinction. But by including the separate times in the same analysis, it erodes any possibility of two taxa truly occupying the same morphological spaces. This could be explored (I am not suggesting the authors perform this analysis, but am moreso representing an example of how this could be performed) in a PCoA scenario through the concept of an n-dimensional hilbert space, by trying to find a plane that passes through the n-dimensional ellipsoid that intersects the clustering of the Bithynian vs the Pelsonian (assuming that the ellipsoid that contains all of the data also contains the clusters of subsets of the full dataset). However, because the nMDS is an abstraction of the data based on the number of desired axes, and not a representation of a mathematical solution, there can be no plane that connects two separate nMDS planes to the 'parent-analysis.' If you want to represent the clustering of points in the same plot, I suggest just two plots, D+J, and use differently colored convex-hulls to outline the subdivisions of the Early-Middle Triassic. I think that this approach would be a stronger representation of your results and would still be readily understandable and accessible.

SUPPLEMENT COMMENTS

Supplementary Figure 11 - The distribution of skull length here makes me suspicious that this plot (and others like it), may be suffering from the Horseshoe effect [5]. It appears that your Nothosauroids and Pistosauroids are generally larger and the Pachypleurosauroids are smaller. It is extremely common for linear measurements that exist on a gradient (body size is classically a problem in this exact scenario), then the ordination essentially bends that straight line into a horseshoe, to maximize the variance. This can make interpretation exceedingly difficult, because a space in the middle would be interpreted as an unfilled ecospace or niche, but if it is just a representation of body size, then we would never expect anything to exist within that space, as it would not exist (because the true variation is actually unidimensional).

Supplementary Figure 11 & 15 - I think replacing these figures with the PCoA results would be more in line with a true ecomorphospace, as the nMDS struggles with issues of reproducibility and abstraction of data that I discuss at length. This change would be in line with both the paleontological literature [3-4], and the literature regarding the description and mathematical explanation for the adaptive landscape of evolutionary quantitative genetics [1-2].

R code, by line:

First and foremost, thank you for providing the R code for this. My comments for this supplementary file are aimed at trying to make sure that even an R-novice can open this file and run these analyses, or easily alter them to fit their data. I do not believe this is currently reasonable, and so below I discuss some of the issues that I ran into, and what I think potential remedies to that can be.

1: The R file is rather difficult to review. I note a major issue below, but a core issue is that you are including everything that you did. That is generally good, but this is 1500 lines of code with not enough comments or documentation of what the individual functions are doing. I had a very difficult time trying to find the code that pertains to which figure, and especially trying to find where in the code the phylomorphospace was actually estimated (I see the section, but there is nothing to note which line of code is generating the adaptive landscape and creating those nested circles, and I was unable to find any R documentation of the ggphylomorphospace function that defined how the landscape was generated). I would recommend uploading a separate file, where an example is shown for each of the major analyses performed, and it is heavily commented (make sure the reader has an extremely hard time getting lost). You can then offer a brief example below each major example that says something along the lines of:

#If we want to analyze patterns for the craniodental region only and not on the whole body, we would change all instances of OBJECT_NAME_WHOLEBODY into OBJECT_NAME_CRANIODENTAL, and change the name of our model objects from OBJECT_WB to OBJECT_CD.

This way, you can give the reader the full experience of how to run the code, but then they don't have to dig and search for each major analysis and try to figure out which line of code aligns with which characteristics in the plots.

56: The code and the attached data are not compatible with one another. The data are submitted as a .xlsx, but the code calls for 6 independent .csv files, which seem to correspond to the to the sheets presented in the .xlsx file. The code and the associated data need to be able to run on any computer, after just installing the necessary packages and potentially changing a working directory. As it stands, a reader would need to go in, and either amend the code to explore .xlsx files, or save each sheet in the .xlsx file as an individual .csv file, so they can do the analyses listed here. This is overly laborious, and the data need to be presented in a way such that they can be analyzed by people who are not comfortable in the R statistical environment, which as a vertebrate paleontology paper, is going to be most of the target audience of this work. If you would like, I include a brief suite of code that can read in .xlsx files, and then create individual objects for each sheet, named after the sheet in the file, to help circumvent this issue. You do not need to use it, but making these data more accessible is an absolute must.

::::: CODE START :::::

library(readxl)
library(tidyverse)

path <- "C:\\Users\\Repository\\File.xlsx"

all_tabs <- path %>%
excel_sheets() %>%
set_names() %>%
map(read_excel,
path = path)

for(i in 1:length(all_tabs)){
assign(names(all_tabs)[], all_tabs[])
}

::::: CODE END :::::

607 - 608: the nexus files associated with this are not included in the supplement. The authors either need to justify why they aren't including those files, or provide them in the final supplement.

REFERENCES
[1] Arnold, Stevan J., Michael E. Pfrender, and Adam G. Jones. "The adaptive landscape as a conceptual bridge between micro-and macroevolution." Microevolution rate, pattern, process (2001): 9-32.

[2] Arnold, Stevan J., et al. "Understanding the evolution and stability of the G‐matrix." Evolution 62.10 (2008): 2451-2461.

[3] Dickson, Blake V., and Stephanie E. Pierce. "Functional performance of turtle humerus shape across an ecological adaptive landscape." Evolution 73.6 (2019): 1265-1277.

[4] Dickson, Blake V., et al. "Functional adaptive landscapes predict terrestrial capacity at the origin of limbs." Nature 589.7841 (2021): 242-245.

[5] Podani, Janos, and Istvan Miklós. "Resemblance coefficients and the horseshoe effect in principal coordinates analysis." Ecology 83.12 (2002): 3331-3343.

[6] Uyeda, Josef C., and Luke J. Harmon. "A novel Bayesian method for inferring and interpreting the dynamics of adaptive landscapes from phylogenetic comparative data." Systematic biology 63.6 (2014): 902-918.

P.S.

Potentially interesting analysis to perform (for a future project)?

I do not expect this to be incorporated into this manuscript, as it would be too time and labor intensive and outside the scope of this particular work, but wanted to note an idea that may be in line with the authors interests, and could be an interesting future problem to explore. The idea of convergence, in all of biology, is classically difficult to quantify and truly represent and the authors do a good job here with these metrics. Another potential approach is to use the time-calibrated phylogeny and look at character evolution models of the various ratios, to establish whether or not different eosauropterygian groups exist on distinct adaptive peaks, and if there are any major transitions. Specifically, I would offer the Bayou package [6]. This package is designed to predict when, and to what effect size, evolutionary transitions between adaptive zones occur. In essence, it is a package that utilizes brownian motion, and Ornstein-Uhlenbeck character evolution models, and is built specifically to work alongside allometric data (which I believe your data would fit perfectly into). Essentially you would score your taxa as following regime 1 or another (geographic division between two clades as delineated in the paper would be perfect for this), and then you attach the linear measurements to the tips of your phylogeny, and it will essentially do an ancestral state estimation, but evaluate 'how many regimes are present?' 'Does this character evolve under brownian motion (null)?' or 'Do we suspect that this feature may have been under some selective pressure (OU)?' This approach is also not subject to the same issue I outline above of species being unable to fill the same niche. Two taxa can exist at the same optimum in this analysis, and independent shifts towards a new optimum can be better evaluated. You can find an example of Bayou being applied to a Permian-Triassic lineage with a time-calibrated (not tip-dated) phylogeny here (https://pubs.geoscienceworld.org/paleobiol/article-abstract/45/1/201/570557/The-evolution-of-the-dicynodont-sacrum-constraint).

·

Basic reporting

-

Experimental design

-

Validity of the findings

-

Additional comments

This is a truly interesting manuscript that provides very relevant information about the diversification, niche partition and ecomorphological adaptations of the early members of Eosauropterygia. It is well written and structured, with clear explanations and a detailed methodology. In the same way, the results are solid and the discussion and conclussions well justified. To my knowledge the analyses are robust, and the figures are informative.
I wish that the manuscript would put more emphasis on the pistosauroid implications, but the data does not allowed a more detailed analysis and discussion. In the attached document I've included some minor comments and corrected some typos. In addition, I would suggest a small correction of the supplementary figure 2, due to Simosaurus can only be be unequivocally identified in Ladinian levels, whereas other remains from the Carnian only allow their identification as indeterminate simosaurids.
Therefore, I would recommend the publication of this manuscript witheminor corrections.

---

## Round 0.2 · accepted · Accept

Thank you for carefully preparing a revised manuscript that incorporates all reviewer's critiques and my own. The R Markdown and the additional analysis using Castiglione et al's methods are a nice addition.

I'm glad to recommend the manuscript for publication as is.

·

Basic reporting

Brenen Wynd comments:

The authors have responded to each of the comments sufficiently and as such this manuscript should be published.

I am not suggesting changes, but I would like to respond to a few of the responses, simply to further the conversation and try to remedy any miscommunications on my part.

"We respectfully disagree that nMDS and PCoA represent data this differently; multiple studies have indeed used nMDS for visualisation (our R2 scores are good enough to do so); but we use PCoA for all calculations. We can – and we now – use the PCoA directly for morphospace visualisation as suggested but the argument against the use nMDS appears too strong in our opinion."

I greatly appreciate this comment. I was never trying to say that nMDS wasn't appropriate for data visualization, it was the concern of using a stochastic method to interpret as macroevolutionary landscapes, whereas a deterministic approach will always produce a plot with the same ordination space and thus the same sized and placed contour-lines.

"Our linear data is composed of ratios that eliminate the direct effect of size. Moreover, we have a series of large and small taxa occupying similar portion of the morphospace. Here, our result appears much more strongly affected by phylogeny, not size."

Ratios implicitly contain information about size as any ratios produced from the same comparison will have the same denominator as one another (body size), and can drive variation in the dataset. The horseshoe effect is an issue for this exact scenario, when the data have a lot of autocorrelation (e.g., DNA), these analyses will tend to bend the data into a horseshoe shape to maximize variance based on body size, even if it isn't explicitly modeled in the dataset.

With that being said, I agree with your assessment given the association of small and large taxa.

An example of how ratios composed via a single common denominator (e.g., body size) impact multivariate analyses is discussed in

( Atchley, W. R., C. T. Gaskins, and D. Anderson. 1976. Statistical properties of ratios. I. Empirical results. Systematic zoology, 25 (2):137-148)

and as that paper notes, your eigenvalues show a marked decrease from the 1st to the 2nd. I'm not saying this should have an impact on your paper, but these are issues that we certainly have to consider when doing these types of analyses.

Experimental design

no comment

Validity of the findings

No comment

Additional comments

No comment

·

Basic reporting

After including the mayor changes suggested by Reviewwer 1 and the Editor, as well as the minor changes I suggested (in the main text, supplemental material and figures), I would accept the manuscript in its current form.

Experimental design

-

Validity of the findings

-

Additional comments

-